# Genome-Wide Exploration of Long Non-Coding RNAs of *Helicoverpa armigera* in Response to Pyrethroid Insecticide Resistance

**DOI:** 10.3390/insects15030146

**Published:** 2024-02-21

**Authors:** Md-Mafizur Rahman, Celso Omoto, Juil Kim

**Affiliations:** 1Agriculture and Life Science Research Institute, Kangwon National University, Chuncheon 24341, Republic of Korea; mmrahman@btge.iu.ac.bd; 2Department Biotechnology and Genetic Engineering, Faculty of Biological Science, Islamic University, Kushtia 7003, Bangladesh; 3Department of Entomology and Acarology, Luiz de Queiroz College of Agriculture (ESALQ), University of Sao Paulo, Piracicaba 13418-900, Brazil; celso.omoto@usp.br; 4Department of Plant Medicine, Division of Bio-Resource Sciences, College of Agriculture and Life Science, Kangwon National University, Chuncheon 24341, Republic of Korea

**Keywords:** *Helicoverpa armigera*, insecticide resistance, lncRNAs, cytochrome P450, RNAseq

## Abstract

**Simple Summary:**

Long non-coding RNAs (lncRNAs) are regulatory molecules involved in various biological processes in *Helicoverpa armigera* (Hübner) (Lepidoptera: Noctuidae). Although research on insect lncRNA is ongoing, research findings associated with lncRNAs are still in the preliminary stages. Therefore, we putatively identified pyrethroid insecticide resistance-related lncRNAs at the genome level. Further, we determined their expression levels in three—low, moderate, and high—pyrethroid insecticide-resistant and -susceptible strains. Notably, cytochrome P450-associated lncRNA expression levels were significantly higher, whereas cuticle protein-related lncRNA expression levels were significantly lower in all susceptible strains than in resistant strains. Further in-depth research should be conducted on the regulatory mechanisms of overexpressed P450 genes as well as their relationship with pyrethroid resistance mechanisms involving lncRNAs in *H. armigera*. Our study provides valuable information for understanding the resistance mechanisms and may help in managing the insecticide resistance of *H. armigera*.

**Abstract:**

Genome-wide long non-coding RNAs (lncRNAs) in low, moderate, and high pyrethroid insecticide-resistant and -susceptible strains of *Helicoverpa armigera* were identified in this study. Using 45 illumina-based RNA-sequencing datasets, 8394 lncRNAs were identified. In addition, a sublethal dose of deltamethrin was administered to a Korean-resistant strain (Kor-T). The average length of lncRNAs was approximately 531 bp, and the expression ratio of lncRNAs was 28% of the total RNA. The identified lncRNAs were divided into six categories—intronic, intergenic, sense, antisense, cis-RNA, and trans-RNA—based on their location and mechanism of action. Intergenic and intronic lncRNA transcripts were the most abundant (38% and 33%, respectively). Further, 828 detoxification-related lncRNAs were selected using the Gene Ontology analysis. The cytochrome P450-related lncRNA expression levels were significantly higher in susceptible strains than in resistant strains. In contrast, cuticle protein-related lncRNA expression levels were significantly higher in all resistant strains than in susceptible strains. Our findings suggest that certain lncRNAs contribute to the downregulation of insecticide resistance-related P450 genes in susceptible strains, whereas other lncRNAs may be involved in the overexpression of cuticle protein genes, potentially affecting the pyrethroid resistance mechanism.

## 1. Introduction

Insect genomes encode a large number of non-coding RNAs that interact with and regulate gene expression and influence insect phenotypes [1,2,3,4]. Non-coding RNAs (ncRNAs), especially long non-coding RNAs (lncRNAs), widely regulate gene expression at the transcriptional level [5,6,7,8]. Elucidating the regulatory functions of lncRNAs may provide insights into insecticide mechanisms, such as the functions of the detoxifying enzyme P450, about which little is known [9]. ncRNAs with molecular sizes >200 and <200 bp are classified as lncRNAs and small ncRNAs, respectively. Notably, both of these ncRNAs are transcribed by RNA polymerase II [6,10]. The arbitrary lengths of ncRNAs are relatively subjective and depend on the researcher involved. Some researchers have considered ncRNAs longer than 200 bp as lncRNAs in several insect genome types [4,11], whereas others have considered ncRNAs longer than 300 nucleotides as lncRNAs [11]. lncRNAs participate in a wide range of gene expression regulatory functions at the transcriptional level [5,6,7,12], providing insights into insecticide mechanisms [1,9]. Recently, lncRNAs have been identified on a genome-wide scale in several insect species [11,13,14,15,16,17]. Notably, 1119, 1309, 3463, and 11,810 lncRNAs have been identified in *Drosophila melanogaster*, *Plutella xylostella*, *Helicoverpa armigera,* and *Bombyx mori*, respectively [11,13,14,17,18].

lncRNAs play significant roles in numerous biological processes [19,20]. The putative functions of lncRNAs can be determined based on their genome locations, co-expressed mRNAs, interactions with transcript sequences, or partial alignment [21]. Furthermore, lncRNAs can regulate either enhancers or suppressors, which can help explain the mechanisms of resistance-related genes in insects [3,12]. However, the relationship between lncRNAs and coding RNAs and their involvement in the insecticide resistance mechanism of *H. armigera* remains unknown [8]. Therefore, investigating the relationship between lncRNAs and coding RNAs in *H. armigera* is crucial. 

Determining the functions of lncRNAs is more challenging than that of protein-coding RNAs. Although protein-coding RNAs can be easily characterized using tools such as BLAST or domain searching in protein databases [22], only a small fraction of the total lncRNA transcripts have been functionally characterized to date [23]. lncRNAs are generally expressed at lower levels and are more tissue-specific than protein-coding RNAs [24,25]. Most lncRNAs have high rates of evolutionary turnover, both in terms of sequence and expression levels, but the tissue specificity of lncRNAs tends to be conserved [26]. However, lncRNAs are important for exploring the regulation of protein-coding genes to understand insecticide resistance mechanisms. The first study of lncRNAs related to insecticide resistance in *P. xylostella* could not functionally characterize the specific insecticide resistance-associated lncRNAs, which showed differences in expression patterns between insecticide-resistant and -susceptible strains [4,11,27]. Further, specific lncRNAs could interact with detoxification enzymes, particularly cytochrome P450, such as CYP6B6, in chlorantraniliprole-resistant strains [4]. 

Cytochrome P450 enzymes encoded by a gene superfamily play crucial roles in the metabolism of both endogenous and exogenous substances in various organisms [28]. They contribute to the resistance of pests to a wide range of insecticides. Multiple overexpressed and deltamethrin insecticide resistance-related cytochrome P450 genes have been identified in the insect pest *H. armigera* [29,30]. Some of these genes are CYP6B7 [31], CYP9A12, CYP9A14, and CYP337B3 [32]. Additionally, cytochrome P450 genes in the Korean *H. armigera* strain were identified in our recently published research, which included the CYP3 subfamily genes CYP337B1, CYP337B2, and CYP337B3, as well as five variants of CYP321A1v1-v5, a newly discovered gene associated with pyrethroid resistance [33]. However, the expression levels of these specific genes can vary among different strains of *H. armigera*, and the contribution of a particular CYP gene to resistance can also differ.

In certain field-derived resistant populations of *H. armigera* in China, fenvalerate resistance was unrelated to CYP337B3 [34]. This suggests that cytochrome P450-mediated resistance in *H. armigera* is a complex phenomenon involving multiple contributing factors. Previous research has also indicated that cytochrome P450, such as chimeric P450 (CYP337B3), is associated with insecticide resistance in *H. armigera* [35]. The presence of chimeric P450 (CYP337B3), resulting from unequal crossing over between CYP337B2 and CYP337B1, has been identified in *H. armigera* populations resistant to fenvalerate in Australia [35], cypermethrin resistance in Pakistan [36], deltamethrin resistance in Korea [37], and many field populations in Brazil [38], as well as worldwide [39]. Previous transcriptomic studies in other insects have suggested that lncRNAs may play a role in susceptibility to pyrethroid insecticides [40,41].

This study focused on identifying novel lncRNAs from the genome sequence of *H. armigera*, a polyphagous species known for its disruptive effects on various crops [42]. This species has developed metabolic resistance to various insecticides, particularly pyrethroids, making it a significant case of insecticide resistance globally [32,36,39]. Metabolic resistance may involve the regulation of the gene expression of specific detoxification enzymes, such as cytochrome P450 [30,33,43] and carboxylesterase [35,44]. The exact mechanisms of action of the lncRNAs in *H. armigera* are still under investigation. Therefore, we investigated lncRNAs from one susceptible and three resistant strains of *H. armigera* using next-generation sequencing (NGS) data. Our findings indicate that a complex regulatory process is involved in resistant strains of *H. armigera*. This study also identified putative pyrethroid insecticide resistance-related lncRNAs at the genome level and their expression levels in one susceptible and three resistant strains.

## 2. Materials and Methods

### 2.1. Insects

Detailed information about the strains of *H. armigera* used in this study, including their origins and resistance levels to pyrethroid insecticides, is presented in our recently published paper [33]. Susceptible (Australian susceptible, TWB-S) and three pyrethroid insecticide-resistant strains (Australian resistant, TWB-R; Korean resistant, Kor-R; and Brazilian resistant strain, bA43) of *H. armigera* were used in this study. The TWB strain originated in Toowoomba, Queensland, Australia, in 2003 and has been maintained in the laboratory. This susceptible strain lacks the CYP337B3 gene and is maintained without insecticide exposure in the laboratory [32]. The three levels of resistance strains used were: (A) The low-resistant (TWB-R) strain, an isogeneic strain of TWB-S, which requires approximately 40 times higher lethal dose (LD_50_) against fenvalerate [32]. This strain harbors the cytochrome CYP337B3 gene, which has been segregated from an identical population. (B) The moderately resistant Korean (Kor-R) strain (collected from a cornfield in Pyeongchang, South Korea, 2013) exhibits moderate but higher resistance to deltamethrin than that of TWB-R [37]. The LC_50_ value of Kor-R for deltamethrin was 110 ppm. Kor-R shows approximately 2503-fold (resistance ratio [RR] = 2503; 110.132/0.044; here, LD_50_ = 0.044 µg/larvae) higher resistance than TWB-S against pyrethroid; however, resistance could not be directly compared with TWB-S [30]. (C) The highly resistant (bA43) strain (collected from soybean fields in Lous Eduardo Magalhaes, Brazil) exhibits approximately 20,000-fold higher resistance against pyrethroids [38]. As the bA43 strain exhibits an extremely high level of resistance to pyrethroids such as deltamethrin, it was difficult to determine the 100% mortality dose or concentration; therefore, the LD_50_ or LC_50_ values could not be calculated. All susceptible and resistant strains (refer to previously described A, B, and C based on the resistance levels) were reared on a Bio-Serv artificial diet (F9772) at 26 °C and 55% relative humidity with a 16:8 (light:dark) photoperiod under the same conditions as previously reported [30]. 

### 2.2. RNA Sample Preparation and RNA-Sequencing (RNA-Seq) Data

RNA was extracted from the different strains of *H. armigera*, including susceptible (TWB-S) and resistant strains (TWB-R, Kor-R, and bA43), as well as a sublethal dose-treated Korean strain (Kor-T). Five larvae (5th instar larvae within 12 h after molting) of each strain were collected in a tube and considered as one sample. Three biological replicates of each sample were used for the RNA-seq analysis. Three different tissue parts were harvested from each insect: the fatbody, gut, and rest of the body. The same replication strategy was used for each sample population. For RNA-seq, four strains and a total of five insect samples were used; the treatment group of one strain (Kor-R) was divided into three tissues, and RNA was extracted using three replicates. Finally, a total of 45 RNA samples were prepared. The Kor-R strain (the treatment strain designated as “Kor-T”) was treated with deltamethrin (sub-lethal dose of 0.002 ng/larva) 24 h before dissection. A sublethal insecticide treatment refers to exposing organisms to an insecticide concentration that does not cause immediate death but has measurable effects on their physiology, behavior, or other biological processes.

Based on the manufacturer’s protocol, 45 RNA-seq data samples were extracted using an RNeasy Mini Kit (Qiagen, Hilden, Germany). The integrity of total RNA, mRNA, and cDNA was validated and quantified using a 2100 Bioanalyzer (Agilent Technologies, Santa Clara, CA, USA). Libraries were prepared using the TruSeq RNA Sample Prep Kit v2 (Illumina Inc., San Diego, CA, USA) and sequenced on the Hiseq4000 platform with the TruSeq 3000/4000 SBS Kit v3 (Macrogen, Seoul, Korea).

### 2.3. lncRNA Identification Pipeline

A lncRNA identification pipeline was developed to discard transcripts based on size and evidence of protein-coding potential. A pipeline for *H. armigera* lncRNA identification is shown in Figure 1. From a total of 73 Gb RNA-seq data sets, 36720 primary transcripts were identified. Of these, 25007 were considered coding transcripts because they were either translated into proteins containing >100 amino acids or their lengths were <200 bp. Another 1886 transcripts were considered coding transcripts after performing a BLASTx search using the SwissProt (cut-off E-value < 0.001) and Pfam v31.0 (cut-off E-value < 0.001) databases [25,40]. An additional 1103 transcripts were discarded after matching with the databases (National Center for Biotechnology Information non-redundant databank, NCBI, NR-DB, and *Drosophila melanogaster* databank, DB). The significant lncRNA-DB in insects was set based on coding potential analysis using the Coding Potential Calculator (cut-off E-value ≥ −1.0) [45] and Coding Potential Assessment Tool (CPAT) v1.2.4 (cut-off value ≤ 0.39) [46]. Housekeeping RNAs (tRNA, rRNA, snRNAs, and snoRNAs) (cut-off E-value < 1.0 × 10^− 10^) were mapped. In addition, 326 transcripts were compared and completely matched to the reference gene (Harm_1.0) [47]. Thereafter, the matched sequences of the reference gene (Harm_1.0) were discarded. Subsequently, the four identified isoform transcripts were discarded as the final lncRNA candidate transcripts. Ultimately, the remaining 8394 transcripts were considered the final lncRNAs in *H. armigera*.

### 2.4. Gene Ontology (GO) Analysis of lncRNA for Functional Annotation

GO analysis was performed using Blast2GO (v5.0) with BLAST (BLASTN program). However, the BLASTN results for the lncRNAs showed sequence similarity to the coding RNA. 

### 2.5. lncRNA Expression Analysis

RNA-seq data were trimmed using Trimmomatic v0.36 [48]. Trimmed high-quality RNA-Seq reads were mapped to the gene sequences, and the TPM and FPKM counts were calculated based on RNA-Seq data on the gene sequences using the RNA-Seq by Expectation Maximization (RSEM) program v1.2.9 [49] with the default parameters. Transcripts with a *p-*value of <0.05 and |log2 (fold change)| > 2 were considered significantly differentially expressed. Statistical analysis was performed based on FPKM values. Additional contig assembly, sequence comparison, and alignment were performed using the Lasergene software v14 (DNASTAR). Trimmed and assembled RNA-seq data were mapped to two reference genomes. The first was Harm_1.0 (isolate: Harm_GR_Male_#8, GCA_002156985.1), which was downloaded from GenBank (http://www.insect-genome.com/waspbase/download/genome_message.php?species=Helicoverpa%20armigera; accessed on 20 June 2023). The second was obtained from the lab genome database, ASM1716586v1 (isolate: CBW_Kor-R33, GCA_017165865.1). To compare expression patterns among strains or tissues, an MA plot [49] was generated using a heatmap function in the R package “gplots” v3.1.1.

### 2.6. Identification of lncRNA and Analysis of Detoxification-Related Genes

Metabolic enzymes may be involved in the detoxification of insect resistance mechanisms [50,51]. Detoxification genes were searched from the annotated sequences of protein databases and protein BLAST and compared to those of *H. armigera* (Harm_GR_Male_#8, GCA_002156985.1). Moreover, most lncRNAs cannot code for proteins but can regulate the expression of related coding mRNAs. To search for the potential function of differentially expressed lncRNAs in susceptible and resistant strains of *H. armigera*, the target genes were predicted using an online prediction tool LncTar v.1.0 (standardized free energy < −0.1, a threshold of −0.01 normalized delta G was set because it is the lowest suggested threshold, which could detect all possible lncRNA–mRNA interactions) [45]. GO analysis of lncRNAs often relies on functional analysis of target genes that are potentially regulated by lncRNAs. Prediction of the target genes of lncRNAs is typically performed based on their co-localization or co-expression patterns with protein-coding mRNAs. Potential interactions and regulatory relationships can be inferred by examining genomic proximity or expression correlations between lncRNAs and mRNAs.

### 2.7. Phylogenetic Analysis of Cytochrome P450 and Cuticle Proteins (CPs)

Detoxification-related enzymes P450 and cuticle-related proteins were screened using different bioinformatics tools (differential gene expression (DEG) pathway, MA plot, and GO analysis) and used for phylogenetic analysis. Related lncRNAs were matched to their respective genes using BLASTN analysis.

Phylogenetic analysis of P450 and CP was performed based on the amino acid sequences of the candidate transcripts, which were identified in this study with significant transcription expression data values. Amino acid sequences were aligned using ClustalW v.2.1 [52]. Unrooted trees with cytochrome proteins were constructed using the neighbor-joining and maximum likelihood methods and the Jones–Taylor–Thorton plus gamma-distributed (JTT+G) model using MEGA11 software [53]. A general time-reversible plus gamma-distributed (GTR+G) model was used to construct the phylogenetic tree. Node support was assessed using a bootstrap procedure based on 1000 replicates. Evolutionary distances were computed using the maximum composite likelihood method and are indicated in units of the number of base substitutions per site.

### 2.8. Statistical Analysis

A post hoc test, including Duncan’s multiple range comparison test (*p* < 0.05), was employed to calculate the mean values (mean values followed by different letters (a, b, c, and ab, bc) in the figures demonstrating the tested samples were significantly different). The standard deviation was calculated to determine the average mean and variation in a specific data group. All statistical calculations were performed using SAS v9.4 (SAS 9.4, SAS Institute Inc., Cary, NC, USA). 

## 3. Results

### 3.1. Identification and Characterization of lncRNAs in H. armigera

A total of 45 RNA-seq datasets were obtained for three different tissue samples (fatbody, gut, and rest of the body) of one susceptible and three resistant strains (three biological replicates for each strain) of *H. armigera*. Figure 1 demonstrates the identification scheme for lncRNAs. We identified 8394 lncRNAs in *H. armigera* (Figure 1 and Figure 2A). The average sequence length of the lncRNAs was 531 bp, ranging from 201 to 8379 bp. The total number of 4,460,338 bp accounted for approximately 1% of the entire genome (337 Mb) [46]. In contrast, coding RNAs comprised 19% of the total genome. The average length of coding RNA and lncRNA sequences was 2278 and 531 bp, respectively. The major identified lncRNAs belonged to class u; however, over 70% of the lncRNAs were included in classes “u”(intergenic) and ”i” (intronic), with 3197 (38.1%) and 2783 (33.2%) lncRNAs, respectively, whereas 1416 (16.9%), 818 (9.7%), 177 (2.1%), and 3 (0.04%) lncRNAs were included in classes “p” (polymerase run), “r” (repeat run), “e” (exotic overlap), and x, respectively (Figure 2B). 

The average lncRNA transcripts accounted for 22.86% (8394/36,720) of the total transcripts, although they varied in different tissues and the susceptible and resistant strains. The location of most lncRNAs in the *H. armigera* genome was identified based on BLASTN results in the Harm_1.0 DB. For example, the longest lncRNA, lnc. Harm_4193 was located in *H. armigera* genomic scaffold_111. Similarly, approximately 20% of the lncRNAs were located in a single genomic scaffold. However, variation existed in the number of lncRNAs, ranging from 1 to 890 (Figure 2D). Only two lncRNAs (lnc_Harm_001 and lnc_Harm_7450) did not match the total (*n* = 998) genome scaffolds of *H. armigera* (Figure 2D). Notably, two lncRNAs could not be identified because of the lack of complete chromosome-level genome studies. Genomic characterization of the lncRNA hits/copy numbers revealed that the highest number of hits was 2073 (24.70%, *n* = 2–50 copies), the second highest was 1671 (19.91%, *n* = 1 copy), and the subsequent hits ranged from 1 to approximately 900 (Figure 2D).

Figure 3 presents the differential expression levels (downregulation and upregulation) of coding and non-coding RNAs. A higher expression level was observed in susceptible (S) and highly resistant strains (bA43), whereas a lower expression level was observed in susceptible and low-resistance strains (Kor-R/S and Kor-T/S) (Figure 3A). Additionally, the isogeneic TWB-R strain with low resistance exhibited a low number of differentially expressed lncRNAs. Six different types of lncRNAs (e, i, p, r, u, and x) were expressed at higher levels in the fatbody, followed by the gut and rest of the body of the resistant strains (Kor-R, Kor-T, and bA43), compared to the susceptible strain (TWB-S) (Figure 3B). 

An interactive MA scatter plot (where “M” represents log 2 (ratio) and “A” represents average intensity or average expression level) was used to filter genes between susceptible and resistant strains based on the M, A, and *p*-values and to compare differentially expressed lncRNA transcripts between insecticide and susceptible strains. The results showed that a considerable number of genes were significantly upregulated (*p =* 0.05), as presented in red dots in Figure 4.

### 3.2. Differentially Expressed lncRNAs and Their Putative Functional Analysis

Differential expression analysis was performed to identify pyrethroid resistance-associated lncRNAs, and GO analysis of the detoxification-related lncRNAs was performed between susceptible and resistant strains of three tissues. Notably, 828 lncRNAs that may be involved in the cellular detoxification process were screened out (Figure 5). The lncRNA expression levels were higher in the fatbody than in the rest of the body and gut tissues of all tested resistant strains (Figure 5).

The cytochrome P450-associated lncRNA expression level was higher in all three tissues from the insects of susceptible strains (TWB-S) than in the resistant strains (TWB-R, Kor-R, Kor-T, and bA43) (Figure 5). Furthermore, we observed higher expression in the fatbody tissue than in the gut tissue and rest of the body of the susceptible and resistant strains (TWB-S > TWB-R). However, no significant differences were observed between the Australian susceptible (TWB-S) and resistant (TWB-R) strains. Using the whole RNA-seq data analysis, 8394 lncRNAs were identified in *H. armigera*; among them, six lncRNAs were highly expressed in *H. armigera* cells. The sequential expression of six lncRNAs (lnc_harm4945 > lnc_harm4755 > lnc_harm 6021) indicated their potential roles in resistance mechanisms. The overall order of the expression levels in the susceptible and resistant strains was as follows: TWB-S > TWB-R > Kor-T > Kor-R > bA43. The relationship between the CP and lncRNAs was stronger in the susceptible strain, whereas the opposite trend was observed for the cytochrome P450 protein (Figure 5D–I). Additionally, the expression of the six lncRNAs was higher in the highly resistant strain bA43 in the tissues of the rest of the body (Figure 5I).

For phylogenetic analysis, 69 highly differentially expressed cytochrome protein-coding sequences were used. The widely studied CYP337B and CYP321A subfamily genes were grouped together, and other cytochrome genes clustered in a separate clade (Figure 6A). Similarly, 59 transcripts of highly differentially expressed CP-related sequences were used for the phylogenetic analysis. Notably, *H. armigera* cuticle 66-like proteins clustered in separate clades (Figure 6C, clades presented in pink color). Additionally, significant differential expression (based on FPKM values, *p* = 0.05) of the six cuticle-related lncRNAs was observed.

Six cytochrome- and CP-related lncRNAs were screened based on their expression levels. Among these, two lncRNAs (lnc_Harm_4945 and lnc_Harm_8365) were matched to the *H. armigera* truncated cadherin-like protein (BtR) gene, whereas two lncRNAs (lnc_Harm_4755 and lnc_Harm_6021) were matched to the respective genes of the BAC pupae DNA. The remaining two lncRNAs, lnc_Harm_1497 and lnc_Harm_2843, matched with the proline-rich proteins HaeIII subfamily 1-like and cytochrome P450, respectively (Figure 6B). The six cuticle-associated lncRNAs, including lnc-Harm_614 and lnc-Harm_615, may be involved in controlling CP and were found to be overexpressed in resistant strains compared to susceptible strains. These two lncRNAs were aligned to the truncated CP sequences (such as 66-like proteins) of insecticide-resistant strains of *H. armigera* (Figure 6D), whereas three lncRNAs (lnc_Harm_702, lnc_Harm_3221, and lnc_Harm_4280) were matched to their respective genes, including BAC pupae DNA, uncharacterized proteins, and dentin sialophosphoprotein-like proteins. Only one lncRNA (lnc_Harm_0922) was matched to *H. zea* isolate GA-R chromosome 19, which is closely related to *H. armigera* (Figure 6D).

## 4. Discussion

The understanding of lncRNAs in insects is still limited compared to mammals [9]. However, advancements in technologies, such as RNA-seq [54], have facilitated lncRNA profiling in various insect species [4,11,55]. Several known mechanisms are involved in insecticide resistance, including metabolic resistance (detoxification or sequestration of insecticides) [20] and epidermal penetration resistance (cuticle alterations that reduce the rate of insecticide penetration) [56]. This study suggests that certain lncRNAs may regulate the expression of detoxification enzymes, such as P450, and insecticide resistance-related proteins, such as CPs. However, the control mechanisms and expression levels of resistance genes, including lncRNAs, remain insufficiently understood. A previous study reported significantly lower expression levels of lncRNAs (*p* = 0.05) than protein-encoding transcripts in insects [4]. We also observed similar results in this study, along with a lower lncRNA transcript length (log bp) than the protein-coding transcripts in *H. armigera* (Figure 2C).

In our previous study [33], we investigated the resistance to pyrethroid insecticides such as deltamethrin in *H. armigera*. Three hypotheses—the presence of mutations in the target site of deltamethrin, genomic variations between susceptible and resistant strains, and differences in gene expression patterns between resistant strains—were explored; however, no clear connection was established. Further, no mutations associated with pyrethroid resistance in the voltage-gated sodium channel were found in the cDNA or genomic DNA of the resistant strains or field populations [33]. Additionally, the RNA-Seq analysis involved 45 Illumina datasets from three tissue sections (fatbody, gut, and rest of the body). DEG analysis revealed that some detoxification enzyme genes, particularly cytochrome P450 genes, were overexpressed in the resistant strains, supporting their role in resistance.

Transcriptional regulation by trans- and/or cis-factors and copy number variations influence the overexpression of specific CYPs [57,58,59]. A study reported that lncRNAs regulate neighboring protein-coding genes in a cis-and-trans manner in *P. xylostella*, thereby influencing immune-related genes. Notably, strand-specific RNA sequencing was used to investigate the roles of lncRNAs in the fatbody of *P. xylostella* during infection with *Metarhizium anisopliae*. Multiple lncRNAs have been identified as potential precursors of microRNAs, forming a complex involved in immune response and development [60]. In this study, we observed variations in the copy numbers of lncRNAs in the susceptible and resistant strains of *H. armigera* (Figure 1). We observed wide variation in lncRNA hits in *H. armigera*, ranging from 1 to 100 (Figure 2D). In a single species, the number of lncRNAs varies depending on the identification pipeline and/or strain [41]. For instance, several studies [3,11,27,41,61] have reported diverse lncRNAs in the whole genome of the diamondback moth (*P. xylostella*). Physiological, functional, and genetic evidence suggests that multiple P450 genes contribute to deltamethrin resistance [4,20,29]. Several studies have documented overexpression of P450 genes in pyrethroid-resistant strains of *H. armigera* [62,63]. The P450 genes (CYP9A12, CYP9A14, and CYP6B7) and insecticide-degradative cytochrome CYP3 clan are classified into CYP6 and CYP9 [64,65], which are constitutively overexpressed in the fatbody of a fenvalerate-resistant strain compared with the susceptible strain [51]. CYP3 is involved in insecticide metabolism via direct detoxification processes [28]. CYP332A1 was significantly expressed in all six field strains surveyed in China [39]. Similarly, in our study, different cytochrome genes, especially CYP3 family genes, such as CYP3337B3v1, CYP3337B3v2, CYP321A14v1, CYP321A1v1, and CYP321A1v5, appeared to be involved in pyrethroid resistance (Figure 5 and Figure 6). However, Xu et al. observed no positive relationship between resistance levels and the expression levels of cytochrome P450 genes [39]. Similarly, in the present study, an exact causal link between specific P450 genes and lncRNAs in the pyrethroid resistance mechanism of *H. armigera* was not observed. This result demonstrates the complexity of the underlying mechanism of P450 genes in the resistance of *H. armigera*.

P450 is one of the main factors involved in the development of resistance to pyrethroids [32,36,38], and genes other than CYP337B3 are involved in the development of high-level resistance. In addition, the resistance to cypermethrin in Pakistan [29] and to fenvalerate in Australia [32] was reported to be conferred by the enzyme CYP337B3 of *H. armigera*, which may also be involved in the resistant strain of *H. armigera* (Figure 5). P450 confers insecticide resistance to susceptible insects (less expressed) than to resistant insects (overexpressed) [28,61]. Overexpression of P450 genes can occur because of insertions of cis- or trans-acting transposable elements in the promoter region [66]. In addition, P450 gene duplication may be a factor in protein overexpression [50].

lncRNAs are involved in various biological processes in insects, including wing development [67], insecticide resistance [55], and interactions with microRNAs. lncRNAs can also interact with microRNAs, acting as competing endogenous RNAs and modulating the expression of target genes [60,68]. A case study in *Drosophila melanogaster* (fruit fly) demonstrated a specific example in which a long non-coding RNA (*bsAS*) controlled wing development. Notably, this regulation involved modulation of blistered/DSRF isoform usage [67]. Another study showed that lnc15010.10 and lnc3774.2 were highly expressed in the cuticle of a malathion-resistant strain of *H. zea*, indicating their role in malathion resistance [55]. Similarly, in this study, we observed higher expressions of lncRNAs, especially lnc_Harm0614 and lnc_Harm0615, in the most resistant bA43 strain of the rest of the body tissue sections (Figure 5I). However, based on the Cuffcompare classification database, lncRNAs can be categorized into various classes ([69] http://cole-trapnell-lab.github.io/cufflinks/cuffcompare/#transfrag-class-codes, accessed on 5 December 2023). In this study, the identified lncRNAs were classified into six categories: “u” (intergenic), “i” (intronic), “x” (anti-sense), “e” (overlapping exon), “p” (polymerase run), and “r” (repeat run) based on their genomic location and neighboring genes (Figure 2B). Specifically, the “u” category is present in the intergenic of two protein-coding CDS locations. The “i” category transcript exists in a transfer falling entirely within a reference known coding gene. The category “x” transcripts overlap with a known reference protein-coding gene on the opposite strand. The “o” category contains the transcripts partially overlapping with a coding gene on the same genomic regions [4,62].

Xu et al. (2022) observed high expression of the cuticle-encoded protein (CP-63) in a tissue-specific manner [70]. CP-63 plays a significant role in deltamethrin resistance in the mosquito *Culex pipiens pallens*. In addition, the overexpression of two P450 genes (CYP4G16 and CYP4G17) contributes to insecticide resistance through a thicker cuticle or by altering the structural components of the proteins responsible for insecticide resistance [70]. Furthermore, the production of cuticular hydrocarbons is maintained by P450, and the CYP4G subfamily affects insecticide penetration, ultimately contributing to insecticide resistance [28,53,71,72]. Transmission electron microscopy of mosquito legs revealed that thicker leg cuticles are involved in the resistance mechanism [72]. In our study, we observed a slight overexpression of P450 genes, specifically CYP4G15-like, which may contribute to cuticle formation in resistant strains compared to the susceptible strain [36]. The lncRNAs, specifically lnc_Harm-614 and lnc_Harm-615, partially matched the CP-66 gene sequence (Figure 6C). These lncRNAs may play a role in cuticle-related resistance mechanisms. A lncRNA in intron 20 of the cadherin alleles is associated with the transcriptional regulation of cadherin in the pink bollworm *Pectinophora gossypiella* [40]. Notably, two lncRNAs (lnc_Harm-4945 and lnc_Harm-8365) matched the cadherin genes in this study, which may play important roles in regulating the resistance in *H. armigera* (Figure 6B). Interaction between lncRNAs and cadherin is involved in *Bacillus thuringiensis* resistance. In addition, differentially expressed lncRNAs in *H. armigera* regulate metabolic pathways and drug metabolism [17]. The total number of novel lncRNAs (8394 lncRNA transcripts; Figure 2A) in *H. armigera* was higher than those in *Bactrocera dorsalis* (6171 lncRNA transcripts) [40] and *P. xylostella* (3324 lncRNA transcripts) [41]. This implies that the number of identified lncRNAs depends on the sample quality, type of insect, and RNA-seq methods. The number of lncRNAs (8267 novel lncRNAs) in *Tribolium castaneum* is higher than that in other insects, and our results align with the data on lncRNAs of other insects [15]. Additionally, insecticide resistance is mediated by multiple mechanisms, including the overexpression of putative CPs [72]. Nevertheless, resistance involving CPs lowers insecticide penetration in the insect body, which has been reported in insects such as *H. armigera* [71]. Furthermore, the mechanism underlying the interaction between lncRNA–CYPs and –CPs may provide a strategy for managing *H. armigera*. Differentially expressed lncRNAs control coding sequences and may be involved in catalytic pathways. Nevertheless, the screened lncRNAs from the susceptible and resistant strains may be involved in cellular detoxification via cis- or trans-regulatory modes [73,74,75].

A synergistic experiment conducted in a previous study demonstrated that high levels of pyrethroid fenvalerate resistance observed in field strains of *H. armigera* could be significantly inhibited by piperonyl butoxide (PBO) [33]. The experiment involved treating TWB-R and TWB-S larvae with deltamethrin at LD_50_ 0.05 µg and LD_10_ 0.01 µg, both with and without PBO. In TWB-S, mortality did not differ significantly between the PBO-treated and untreated groups at both LD_50_ and LD_10_ doses, suggesting that PBO did not have a significant impact on susceptibility. However, in TWB-R, mortality significantly differed between the PBO-treated and untreated groups, indicating that the inhibition of cytochrome P450 (CYP) enzymes resulted in increased mortality. This consistent pattern was observed across various doses, supporting the conclusion that CYP enzymes play a major role in the development of resistance in TWB-R. Although the contribution of CYP enzymes has been recognized in the development of insecticide resistance, determining the extent of resistance in Kor-R remains challenging [36]. PBO is a widely recognized inhibitor of P450 enzymes. These results further validate the crucial role of P450 enzymes in fenvalerate resistance in *H. armigera*. The fact that PBO effectively reverses resistance suggests that P450 enzymes are involved in the detoxification and metabolism of fenvalerate in *H. armigera* populations [43].

The TWB-R (low-resistant strain) is an isogenic strain of TWB-S (the susceptible strain), which exhibits very high genetic homology. Owing to this genetic homology, a few lncRNAs present in TWB-R may be differentially expressed in TWB-S. Notably, some lncRNAs are commonly differentially expressed in Kor-R, which exhibits a relatively moderate level of resistance, and bA43, which exhibits a high level of resistance, compared with TWB-S and TWB-R, suggesting that lncRNAs may be involved in the development of insecticide resistance. This hypothesis suggests a regulatory mechanism by which the expression of P450 genes is influenced by the expression levels of the associated lncRNAs (Figure 6). In resistant strains, downregulation of these lncRNAs may lead to higher expression of P450 genes, potentially contributing to insecticide resistance. Conversely, in TWB-S, differential regulation or higher expression of P450-associated lncRNAs may play a role in maintaining normal or lower expression levels of P450 genes. In this study, we observed differences in the expression levels between TWB-S and the highly resistant strains (Kor-R and bA43). Furthermore, the isogenic low-resistance strain (TWB-S/R) exhibited a smaller number of lncRNAs compared to the medium- (Kor-R/Kor-T) and high-resistance strains (bA43). This finding is consistent with those of the previous studies of closely related insects, including *P. xylostella* [4], *B. dorsalis* [55], *Spodoptera litura* [76].

This study has limitations in terms of the clear interaction or relationship between specific cytochrome P450 and specific lncRNAs in the pyrethroid resistance mechanism. Experimental validation, including strand-specific RT-PCR or functional studies with RNA interference [40], is crucial for confirming the involvement of lncRNAs in resistance mechanisms. Additionally, cytochrome-related lncRNAs were less distinguishable between susceptible (TWB-S) and resistant (TWB-R) strains at the three different tissue levels. Under stress from various insecticides, CYP genes exhibit tissue-specific expression patterns and shifts in expression levels [77]. Based on previous research and our data analysis, we speculate that certain lncRNAs may be involved in the downregulation of insecticide resistance-related P450 genes in susceptible strains, whereas other lncRNAs may be involved in the overexpression of CP genes, which would ultimately affect the pyrethroid resistance mechanism. The mechanism of resistance is complex and depends on multiple factors in *H. armigera*. Although the lepidopteran-specific CYP6AE subfamily has been previously implicated in pesticide resistance in *H. armigera*, this study did not observe the expression of the CYP6 clan. Notably, pesticide- and xenobiotic-metabolizing P450 genes are predominantly found in the CYP6 and CYP9 families of the CYP3 clan [28]. To date, little is known about the effects of lncRNAs on the regulation of expression of the major detoxifying enzyme P450 [72]. Nevertheless, lncRNAs can act as either activators or repressors in the regulation of gene expression by directly binding to transcription factors or playing a role in DNA methylation. Histone modifications regulate the expression of P450 genes at the transcriptional level, whereas ncRNAs can influence P450 expression at both the transcriptional and post-transcriptional levels [72].

In the present study, we identified pyrethroid insecticide resistance-related lncRNAs in the *H. armigera* genome. A total of 8394 lncRNAs were annotated, 828 detoxification-related lncRNAs were screened, and six putative lncRNAs related to cytochrome P450 and CPs were identified. Nevertheless, further research should be conducted on how lncRNAs regulate resistance mechanisms in insects to better understand the function of each lncRNA.

## 5. Conclusions

In this study, 8394 lncRNAs were identified in susceptible and resistant *H. armigera* strains using 45 RNA-seq datasets following systematic screening criteria. Among these, six lncRNAs were associated with regulatory roles in cuticle- and P450-related proteins. Specifically, low expression levels of certain lncRNAs were associated with decreased expression of P450 genes (CYP337B and CYP321A) in susceptible strains. In contrast, some lncRNAs were overexpressed and associated with CP genes (CP-66). These findings suggest that the expression of specific lncRNAs ultimately affects pyrethroid resistance in *H. armigera*. The current study did not observe a positive correlation between the expression levels of highly expressed cytochrome P450 genes and the three tested resistant strains in the three different tissues analyzed (fatbody, gut, and rest of the body). Further in-depth research should be conducted on the regulatory mechanisms of overexpressed P450 genes as well as their relationship with pyrethroid resistance mechanisms involving lncRNAs in field populations of *H. armigera*. Despite these limitations, this study contributes to the understanding of the role of lncRNAs in the resistance process and lays the foundation for future research on resistance in *H. armigera*.

## Figures and Tables

**Figure 1 insects-15-00146-f001:**
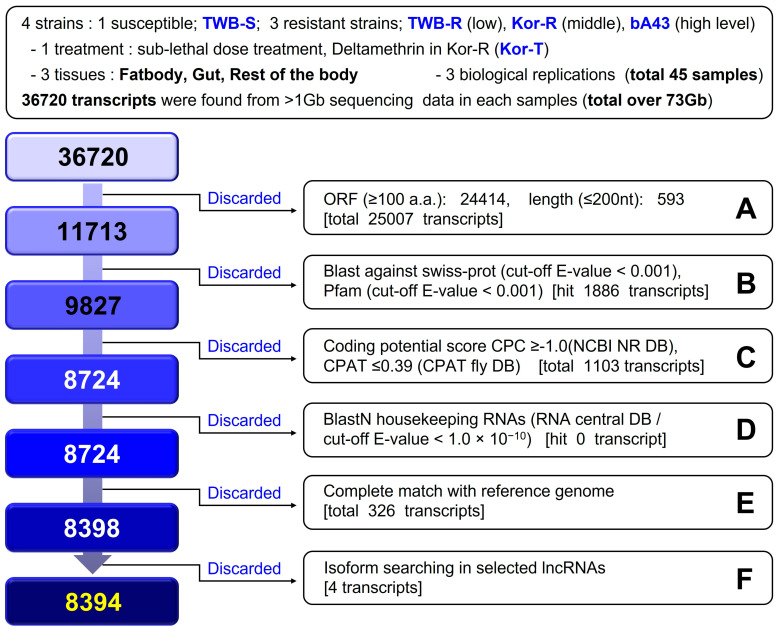
Schematic of identification of long non-coding RNAs (lncRNAs) in *Helicoverpa armigera*. (A) Within the entire 36720 transcripts, a total of 25007 transcripts were excluded, including 24414 transcripts containing ORFs (open reading frames) coding for more than 100 amino acids, and 593 transcripts with short sequences of less than 200 nt in length. (B) Based on the BLAST, 1886 transcripts predicted to encode proteins, that is, excluded by the swiss-prot (cut-off, E-value < 0.001) and Pfam (cut-off, E-value < 0.001) criteria, were excluded. (C) Using the Coding Potential Calculator (cut-off E-value ≥ −1.0) and Coding Potential Assessment Tool (CPAT) (*Drosophila melanogaster* database, fly DB, cut-off value ≤ 0.39) programs, 1103 transcripts outside the range of lncRNAs were excluded. (D) Transcripts corresponding to housekeeping RNAs (RNA central DB/cut-off E-value < 1.0 × 10^−10^) such as rRNA and tRNA were not detected through BlastN. (E) Among the transcript base sequences, 326 transcripts that perfectly matched the base sequence of the reference genome were excluded. (F) Among the selected lncRNAs, 4 transcripts identified as isoforms were excluded, and 8394 lncRNAs were finally selected.

**Figure 2 insects-15-00146-f002:**
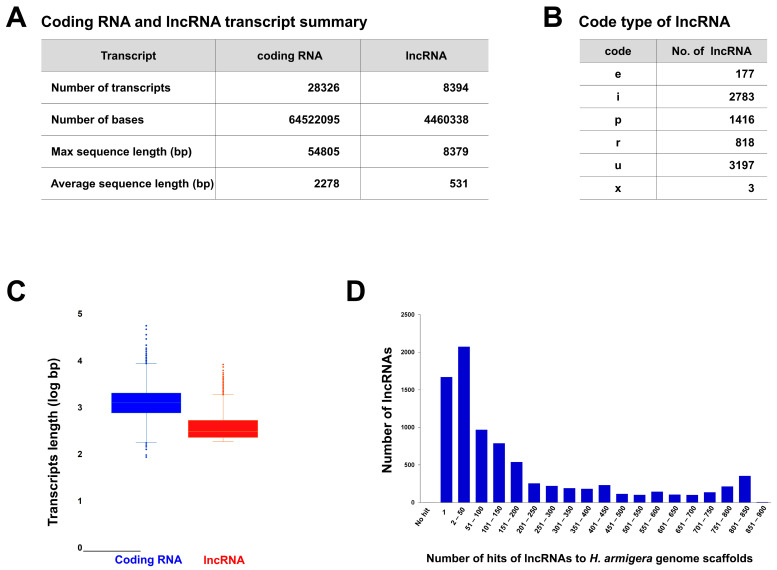
Characteristic analysis of identified lncRNAs from *H. armigera.* (**A**) Comparisons of the number of coding and lncRNA transcripts. (**B**) Code type of lncRNA: e: single exon transferal overlapping a reference exon and at least 10 bp of a reference intron, indicating a possible pre-mRNA fragment; i: a transferal falling entirely within a reference intron; p: possible polymerase run-on fragment (within 2Kb of a reference transcript); r: repeat; u: unknown, intergenic transcript; and x: exotic overlap with reference on the opposite strand. (**C**) Comparisons of transcript length among coding and lncRNAs. (**D**) Number of hits of lncRNAs to the reference genome.

**Figure 3 insects-15-00146-f003:**
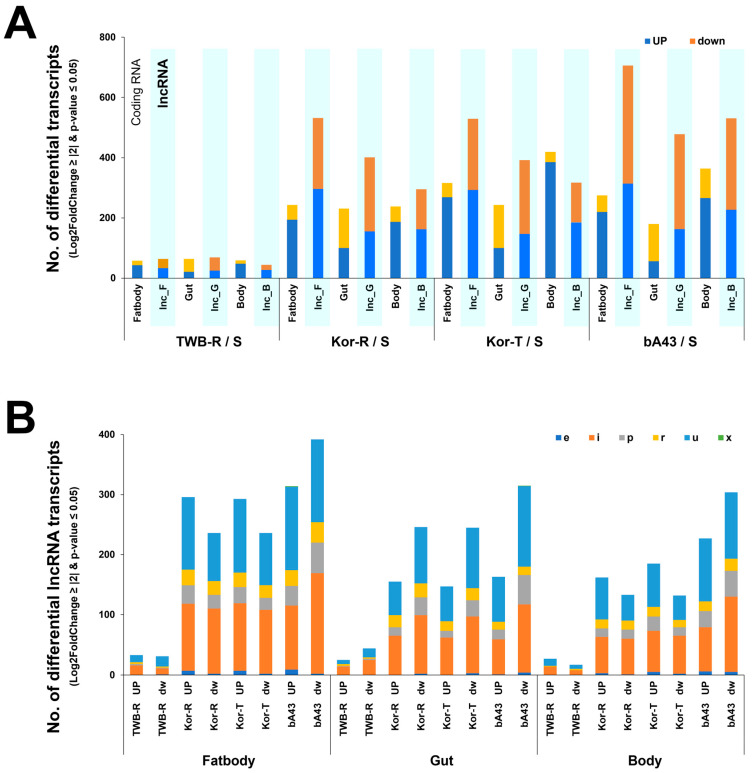
A number of differentially expressed coding and lncRNAs in different strains and tissues. (**A**) Comparisons of differentially expressed transcripts in insecticide-resistant strains (TWB-R, Kor-R, Kor-T, and bA43) compared to those of the susceptible strain (TWB-S) of *H. armigera*. In the *x*-axis, the symbols “Lnc_F,” “lnc_G,” and “lnc_B” indicate the number of lncRNAs in fatbody tissue, gut tissue, and the rest of the body, respectively, in the tested susceptible and resistant strains. (**B**) Comparisons of upregulated or downregulated differentially expressed lncRNAs and their composition in different body tissues of the tested susceptible and resistant strains. Categories of lncRNA types have been described in Figure 2 and in the materials and methods section.

**Figure 4 insects-15-00146-f004:**
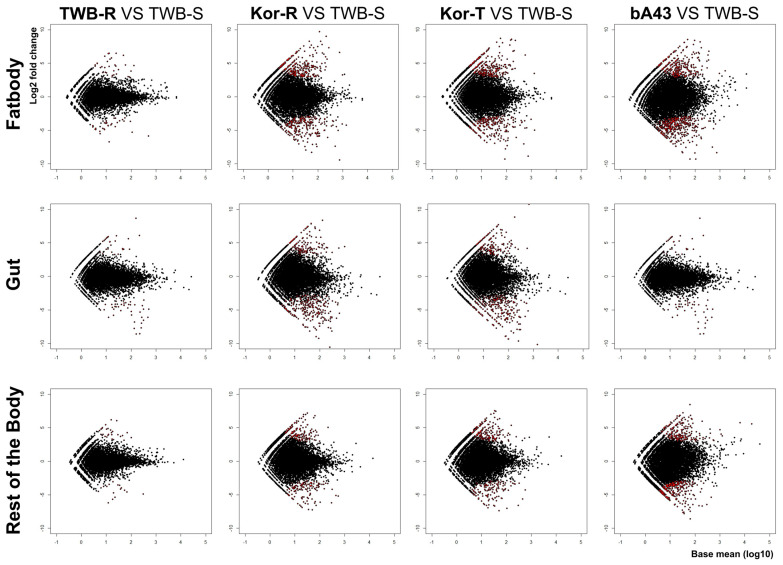
MA plot results based on the RNA-sequencing for differentially expressed lncRNA transcripts from resistant strains compared to the insecticide-susceptible strain, TWB-S. The *y*-axis represents Log2 fold change, the *x*-axis represents the normalized mean expression value, and the *p*-value ≤ 0.05 based on the statistical test is indicated as red dots.

**Figure 5 insects-15-00146-f005:**
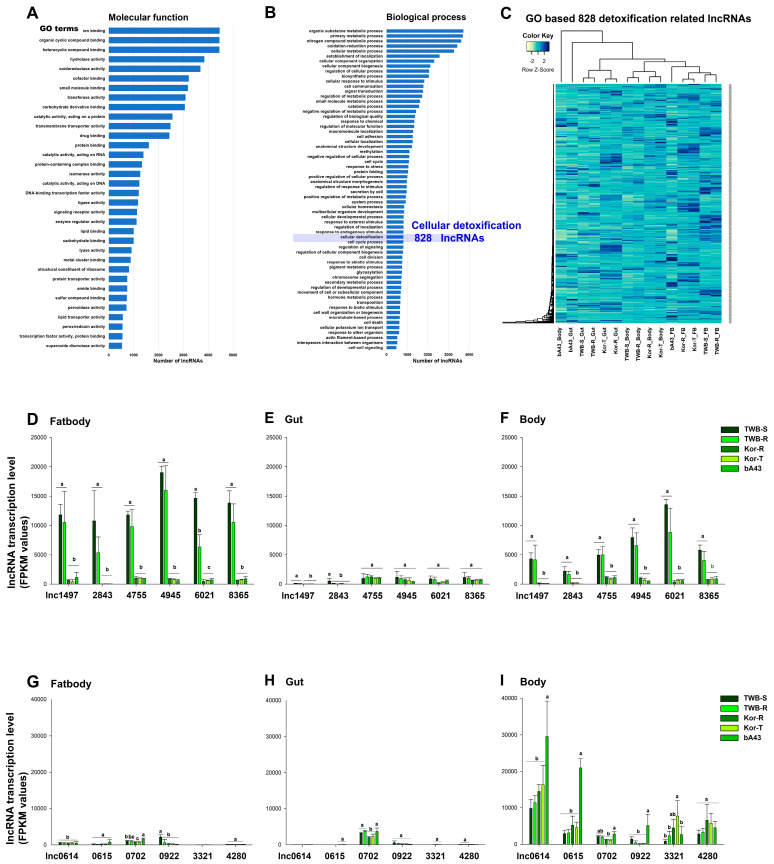
Gene Ontology (GO) analysis results of lncRNAs using Blast2GO (level 3, (**A**) molecular function, and (**B**) biological process). (**C**) Heatmap representing differentially expressed GO-based 828 detoxification-related lncRNAs in insecticide-susceptible strain TWB-S and insecticide-resistant strains TWB-R, Kor-R, Kor-T, and bA43 in the fatbody, gut, and body (rest of the body). (**D**–**F**) Comparison of expression levels between cytochrome P450 enzymes and cuticle proteins (CPs). The lncRNA expression level was higher in all parts of the insect from the susceptible strain (TWB-S) compared with resistant strains (TWB-R, Kor-R, Kor-T, and bA43). (**G**–**I**) LncRNA (especially lnc_Harm0614 and lnc_Harm0615) expressions were higher in the rest of the body of the most resistant bA43 strain. Details on samples are provided in the material and methods section.

**Figure 6 insects-15-00146-f006:**
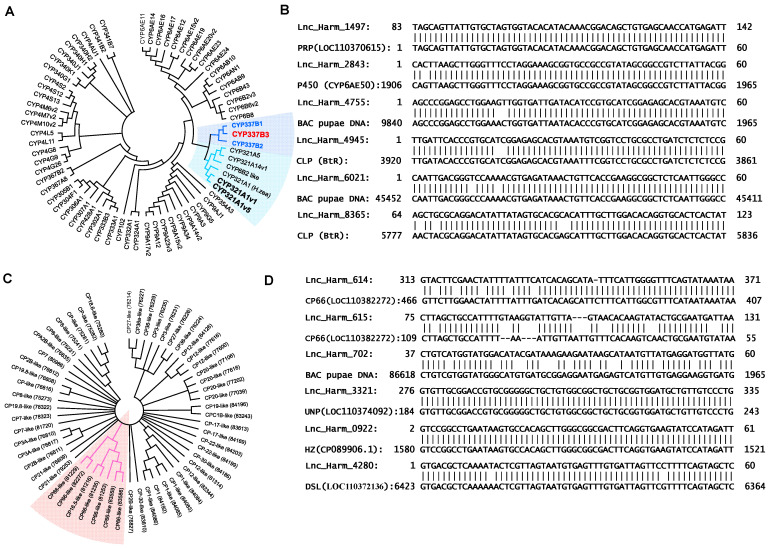
Phylogenetical analysis of cytochrome P450-coding (CYP) transcripts. (**A**) The transcripts of highly differentially expressed CYP proteins (*n* = 69) were used for phylogenetic constructions. The widely studied CYP337B (B1 and B2 marked in violet and B3 marked in red) and CYP321A (v1 and v5 bold) subfamily genes were grouped and shaded in different colors. (**B**) Six significant (based on FPKM values, *p* = 0.05) CYP protein-related lncRNAs were screened out. Among them, two lncRNAs (lnc_Harm_4945 and lnc_Harm_8365) were matched to the *H. armigera* truncated cadherin-like protein (BtR) gene, whereas two lncRNAs (lnc_Harm_4755 and lnc_Harm_6021) were matched to the *H. armigera* BAC pupae. The remaining two lnc_Harm_1497 and lnc_Harm_2843 lncRNAs were matched to the proline-rich protein HaeIII subfamily 1-like and cytochrome P450, respectively. (**C**) Phylogenetic analysis of CP transcripts. The transcripts of highly differentially expressed CP (n = 59) were used for phylogenetic constructions and cuticle 66-like proteins of *H. armigera* clustered in a separate clade are marked in pink. (**D**) Probable binding sites of lncRNAs with CP in *H. armigera.* Six significant (based on FPKM values, *p* = 0.05) cuticle-related lncRNAs were screened out. Among them, two lncRNAs (lnc_Harm_614 and lnc_Harm_615) were matched to the *H. armigera* CP 66-like, whereas the remaining lncRNAs were matched to the other proteins of *H. armigera* except lnc_Harm_0922. PRP, proline-rich protein; CLP, truncated cadherin-like protein; CP, cuticle protein; UNP, uncharacterized protein; HZ, *H. zea*; DSL, dentin sialophosphoprotein-like protein.

## Data Availability

*Helicoverpa armigera* (Kor-R strain) genome assembly was submitted as ASM1716586v1 (contig level) and ASM2626255v1 (chromosome level). The data used in this study are available upon request from the corresponding author.

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
