# Peer review of "Genome-Wide Exploration of Long Non-Coding RNAs of Helicoverpa armigera in Response to Pyrethroid Insecticide Resistance"

_insects, 2024, doi:10.3390/insects15030146_

Round 1

Reviewer 1 Report

Comments and Suggestions for Authors

When I started reading this manuscript, I thought it would be an excellent paper, but after reading the entire manuscript I was disappointed. 

Major comments:

1. First, the methods section is missing description about one entire treatment (sub-lethal treatment of one strain of H. armigera with pyrethroid). I got confused with authors saying 45 seq samples, while there was description provided only for 36. Then while reading the results section, I realized where the total 45 samples came from.

2. Second, in the discussion section, authors talk about suppression of pyrethroid resistance with PBO in either one or some of their strains. However, I did not see the results presented for the PBO study.

3. Third, the authors presented a hypothesis (last 1-3 sentences of the abstract) saying that downregulation of P450-associated lncRNAs in resistant strains could allow overexpression of P450's and vice versa in the susceptible strain. However, this hypothesis or possibility was never discussed in the paper again. 

Minor comments:

1. L33-34: The words intronic and intergenic are mentioned twice in the same sentence. Please rectify for clarity.

2. L51: What authors are trying to say with words "insecticide mechanisms" is not clear. Please provide some context.

3. L56: Is there a specific name for insect ncRNA's longer than 300 nucleotides. It is not clear what authors want to say here.

4. L80-81: The sentence "They have...... and expression [29]" is not clear. It needs to be rephrased for clarity.

5. L118: Delete repeated words.

6. Section 2.2: At least mention the bioassay name (topical, feeding, dipping etc.) in this paragraph. What sub-lethal dose was used to treat one of the three strains for RNAseq experiments?

7. L156-157: No need to mention the collection information in the bioassay paragraph. It is already mentioned in 2.1.

8. Section 2.3. Replication information is presented in an ambiguous manner. Please include following information: (i) How many larvae per replication. (ii) How many replicates per insect population. (iii) Mention whether the replication strategy was same or different for each population.

9. L163: When you say "sample" do you mean to say "replicate"? 

10. L167-168: Three biological replicates of each "insect" or "population"?

11. L191-192: Reference "gene" or "genome". Please correct.

12. Correct the section number for statistical analysis. 

13. L247-249: What type of analysis was performed prior to post-hoc testing? This information is missing.

14. L249-250: The std. deviation bar and "letter" information should be mentioned in figure caption when necessary.

15. L156/57: The information on replicates should be moved to the methods section.

16. L267-268: When authors are talking about expressing on non-coding and coding RNAs, are they referring to their own results or that from a previous study. Clarify this in the paper. Authors should remember that discussing previous work is better suited for the discussion section.

17. Fig. 1. In box E, should "gene" be "genome"?

18. L275: This is the second or third sentence that the authors are talking about size length variation for the non-coding RNAs in the same section. Be sure not to be redundant and keep the manuscript as concise as possible.

19. 334-339: Why are the authors discussing the results here when there is a separate "discussion" section.

20. Figures: Certain figure captions (2, 5 and 6) are too long and can be reduced in length by 50% or more.

Comments on the Quality of English Language

Lines 374-376: There are many grammatical errors throughout the paper. I am pointing out this sentence as an example because its meaning does not make any sense due to grammatical errors.

Author Response

Reviewer1

When I started reading this manuscript, I thought it would be an excellent paper, but after reading the entire manuscript I was disappointed. 

Major comments:

  1. First, the methods section is missing description about one entire treatment (sub-lethal treatment of one strain of H. armigera with pyrethroid). I got confused with authors saying 45 seq samples, while there was a description provided only for 36. Then while reading the results section, I realized where the total 45 samples came from.

Response: Thanks for your comments and suggestions which are helping us to improve the MS. We did not properly description regarding the number of samples. Sample preparation and exact number of samples are summarized again in lines 177-180.

So again, for RNA-seq 4 strains were used, and a total of 5 insect samples including the treatment group of 1 strain (Kor-R), were divided into 3 tissues, RNA was extracted, 3 repetitions, and finally, a total of 45 RNA samples were prepared.

  1. Second, in the discussion section, authors talk about suppression of pyrethroid resistance with PBO in either one or some of their strains. However, I did not see the results presented for the PBO study.

Response: We described that part in a way that made the explanation less clear.

To avoid reader confusion, the bioassay part was deleted from the Materials and Methods section. In addition, the review section was revised to 'previous study' and was revised to cite existing research.

We provided the PBO study in our published manuscript. (Reference No. [36]. Genome-Wide Exploration of Metabolic-Based Pyrethroid Resistance Mechanism in Helicoverpa armigera. bioRxiv 2023, 2023.12.18.572109, (doi:10.1101/2023.12.18.572109).

  1. Third, the authors presented a hypothesis (last 1-3 sentences of the abstract) saying that downregulation of P450-associated lncRNAs in resistant strains could allow overexpression of P450's and vice versa in the susceptible strain. However, this hypothesis or possibility was never discussed in the paper again. 

Response: Many thanks for your valuable comments regarding the study's limitations.  In our study, we mentioned this point in a separate section “limitations” lines 571-593.

There are still not many studies on the functional analysis of lncRNAs in insects, especially lepidopteran insects. Moreover, there are almost none in Heliothine pests, and research on functional analysis is the next step, this study can be said to be the basic and first step of lncRNA research in H. armigera.

Minor comments:

  1. L33-34: The words intronic and intergenic are mentioned twice in the same sentence. Please rectify for clarity.

Response: Updated

  1. L51: What authors are trying to say with words "insecticide mechanisms" is not clear. Please provide some context.

Response: Updated

  1. L56: Is there a specific name for insect ncRNA's longer than 300 nucleotides. It is not clear what authors want to say here.

Response: Updated

  1. L80-81: The sentence "They have...... and expression [29]" is not clear. It needs to be rephrased for clarity.

Response: Updated

  1. L118: Delete repeated words.

Response: Updated

  1. Section 2.2: At least mention the bioassay name (topical, feeding, dipping etc.) in this paragraph. What sub-lethal dose was used to treat one of the three strains for RNAseq experiments?

Response: Updated

  1. L156-157: No need to mention the collection information in the bioassay paragraph. It is already mentioned in 2.1.

Response: Updated

  1. Section 2.3. Replication information is presented in an ambiguous manner. Please include following information: (i) How many larvae per replication. (ii) How many replicates per insect population. (iii) Mention whether the replication strategy was same or different for each population.

Response: Updated

  1. L163: When you say "sample" do you mean to say "replicate"? 

Response: Updated

  1. L167-168: Three biological replicates of each "insect" or "population"?

Response: Updated

  1. L191-192: Reference "gene" or "genome". Please correct.

Response: Updated

  1. Correct the section number for statistical analysis. 

Response: Updated

  1. L247-249: What type of analysis was performed prior to post-hoc testing? This information is missing.

Response: We specify the type of analysis performed before post-hoc testing. We conducted primary statistical analyses, such as ANOVA (Analysis of Variance), to assess the main effects before moving on to post-hoc tests to investigate pairwise group differences (Updated).

  1. L249-250: The std. deviation bar and "letter" information should be mentioned in figure caption when necessary.

Response: Updated.

  1. L156/57: The information on replicates should be moved to the methods section.

Response: Updated.

  1. L267-268: When authors are talking about expressing on non-coding and coding RNAs, are they referring to their own results or that from a previous study. Clarify this in the paper. Authors should remember that discussing previous work is better suited for the discussion section.

Response: Updated.

  1. Fig. 1. In box E, should "gene" be "genome"?

Response: Updated.

  1. L275: This is the second or third sentence that the authors are talking about size length variation for the non-coding RNAs in the same section. Be sure not to be redundant and keep the manuscript as concise as possible.

Response: Updated.

  1. 334-339: Why are the authors discussing the results here when there is a separate "discussion" section.

Response: Updated.

  1. Figures: Certain figure captions (2, 5 and 6) are too long and can be reduced in length by 50% or more.

 Response: Updated.

Comments on the Quality of English Language

Lines 374-376: There are many grammatical errors throughout the paper. I am pointing out this sentence as an example because its meaning does not make any sense due to grammatical errors.

 Response: English Updated (attached certificate of English correction).

Reviewer 2 Report

Comments and Suggestions for Authors

Comments to the manuscript insects-2812649 intended as a research article in Insects entitled “Genome-wide exploration of long non-coding RNAs of Helicoverpa armigera in response to pyrethroid insecticides resistance” by Md-Mafizur Rahman, Celso Omoto, and Juil Kim.

They present an interesting manuscript investigating potential insect resistance mechanisms related to lncRNA in Helicoverpa. A thorough and basic study paving the way for future experimental research to elucidate the involvement of lncRNAs in insecticide resistance.

The manuscript has a comprehensive and good introduction to the topic. The methods used, the number of replicates and the analyses performed seem appropriate and solid.

The results are clear and concise presented. The addition of PBO is mentioned in the method-section and the discussion, but it does not appear in any figure or table. Is that by purpose? Consider including PBO results in figures, tables or text in the result-section or reconsider mentioning PBO in this manuscript.

The comprehensive discussion is covering the findings and topic nicely. Well-structured and well-written. The isogenic TWB-R strain with low resistance have a very low number of differential expressed lncRNA’s. Is this due to resistance status or isogenic status? Please comment and discuss the impact of this and the high number of differential expressed lncRNAs in the other strains.

Figure 5. Not all readers have a magnifying glass. You must change the Y-scale of E, G and H.

Author Response

Reviewer2

Comments to the manuscript insects-2812649 intended as a research article in Insects entitled “Genome-wide exploration of long non-coding RNAs of Helicoverpa armigera in response to pyrethroid insecticides resistance” by Md-Mafizur Rahman, Celso Omoto, and Juil Kim.

They present an interesting manuscript investigating potential insect resistance mechanisms related to lncRNA in Helicoverpa. A thorough and basic study paving the way for future experimental research to elucidate the involvement of lncRNAs in insecticide resistance.

The manuscript has a comprehensive and good introduction to the topic. The methods used, the number of replicates and the analyses performed seem appropriate and solid.

Response: Thanks for your positive comments and suggestions for improving the present manuscript.

The results are clear and concise presented. The addition of PBO is mentioned in the method-section and the discussion, but it does not appear in any figure or table. Is that by purpose? Consider including PBO results in figures, tables or text in the result-section or reconsider mentioning PBO in this manuscript.

Response: We have carefully considered your comments. We provided the PBO study in our published manuscript. (Reference No. [36]. Genome-Wide Exploration of Metabolic-Based Pyrethroid Resistance Mechanism in Helicoverpa armigera. bioRxiv 2023, 2023.12.18.572109, (doi:10.1101/2023.12.18.572109).

The comprehensive discussion is covering the findings and topic nicely. Well-structured and well-written. The isogenic TWB-R strain with low resistance have a very low number of differential expressed lncRNA’s. Is this due to resistance status or isogenic status? Please comment and discuss the impact of this and the high number of differential expressed lncRNAs in the other strains.

Response:  

Yes. You are right. The TWB-R (low resistant strain) is an isogenic strain of TWB-S (susceptible strain) and has very high genetic homology. Because of this genetic homology, TWB-R may be natural that there are few lncRNAs differentially expressed with TWB-S. However, in Kor-R, which has a relatively high level of resistance, and bA43, which has a very high level of resistance, there is some lncRNA that is commonly differentially expressed compared to TWB-S and R, suggesting that lncRNA possibly involved in the development of insecticide resistance.

So we edited Discussion section, Lines 550-567.

Figure 5. Not all readers have a magnifying glass. You must change the Y-scale of E, G and H.

Response: Updated the Figure. 
